# Acceptability of Multiparticulate Dosing Using Sympfiny^®^ Delivery System with Children (Age 1–12)

**DOI:** 10.3390/pharmaceutics14112524

**Published:** 2022-11-19

**Authors:** Kate Abeln, Kate Cox, Laura Haggerty, Mary Beth Privitera

**Affiliations:** HS Design, Morristown, NJ 07960, USA

**Keywords:** multiparticulate, acceptability, pediatric formulation, palatability, oral syringe, administration device, mouthfeel

## Abstract

This study investigated multiparticulate formulation administered over a two-week period of time via the Sympfiny^®^ system with children of ages 1–12 years. The study was conducted with parent–child pairs (N = 120 total participants) following a specific dose strategy to mimic PURIXAN’s dosing guidelines based upon the child’s age. PURIXAN^®^ (mercaptopurine) and Methotrexate have been identified as potential chemotherapy drugs that could benefit from reformulation into multiparticulate. Multiparticulate drugs have advantages as they can be flavorless, and do not require liquid reconstitution and do not require refrigeration. The study included three parts: initial in-person session, 14 days of at-home use, and a final in-person session. The in-person sessions were conducted at HS Design’s (HSD) (Morristown, NJ, USA) offices located in Morristown, New Jersey, where a study moderator captured and recorded all subjective comments by participants and observed device use to identify use errors. The participants were instructed to administer a dose (placebo) for the next 14 days and at each dose delivery to fill out a daily survey regarding their experience. Overall, the cumulative survey responses and feedback collected during the in-person sessions suggest that child participants ages 5–12 years old found multiparticulate to be an acceptable formulation and would be willing to take this medication if they were sick. Over time, more children ages 1–4 did not open their mouths; consistently around 15–20% of 1–4 years olds spat the placebo. However, approximately 95% of parents found the Sympfiny^®^ system acceptable and indicated that they would use it to deliver medication to their child.

## 1. Introduction

Acceptability of a medication is a major component of a drug’s feasibility. As such the primary objective of this paper is to evaluate whether it is acceptable to dose and dispense taste-masked multiparticulate drugs directly to a child using a designed system. Testing for acceptability is difficult due to the lack guidance and fragmented knowledge [1]. The EMA’s veterinary guidance on palatability provides additional guidance on acceptability when testing is conducted on animals. It states that 50 animals should be tested for medications with one dose and 25 animals should be tested for two doses [2]. Of the tested population, 70 percent of the animals (80% if the animals are dogs) will have to voluntarily accept the medication, where voluntary acceptance is defined as “The willingness of the target animal to consume voluntarily and spontaneously the veterinary medicinal product from bowl/trough/ground or from hand when offered as a treat by the animal owner” [2]. The EMA guideline clearly defines requirements of acceptability for medications of veterinary purposes, while “acceptance criteria [for pediatric medications] are left to the applicant” [1].

Though the criterion for medication acceptance is left for the applicant to determine, certain characteristics of acceptability are commonly examined. They include a medication’s: palatability, swallowability, appearance, complexity of modification prior to administration, required dose, dosing frequency and duration of the treatment, administration device, primary and secondary container system, and actual mode of administration [3]. Among these, palatability is the most common characteristic, which is often assessed by observing facial expressions, utilizing scaling methods, and listening to verbal feedback [4].

In the literature, different diary methods (paper, paper with signaling to fill out, and electronic) are used to collect data [5]. While each have been used to record feedback from study participants, they have their own advantages and disadvantages. A study conducted to compare paper and digital diaries found that paper diaries have a higher rate of completing all required entries for a particular day. Whereas participants that use digital diaries may not have completed all the required logs for the day but have a higher rate of completing a daily entry [6]. In addition, digital diaries have higher compliance rates and are found to be superior to paper diaries if time is an important part of the study. Digital diaries allow the study participants the ability to send and set reminders and can record timestamps of when the entry occurred [6].

In a pivotal study, the University of Birmingham utilized a paper questionnaire for an acceptability study that was conducted with the Sympfiny^®^ 2 mL system. A paper questionnaire with several 5-point hedonic scales was provided to 40 child (ages 4–12 years old) participants in order to evaluate different dosage volumes of a multiparticulate (MP) formulation [7]. The children used the 5-point hedonic scale to rate certain qualities about the palatability of the MP formulation [7]. Children as young as 4 years-old were found to be able to competently use a 5-point hedonic scale to describe their perception of a substance or object [8].

The objective of this study was to evaluate the acceptability of the Sympfiny^®^ 1 mL system by adult participants serving as dose administrators and their respective children (ages 1–12) over a 14-day period. Sympfiny^®^ 1 mL is a delivery system intended for oral dosing of oncology medications that are formulated as an MP. PURIXAN^®^ (mercaptopurine) and Methotrexate have been identified as potential chemotherapy drugs that could benefit from reformulation into multiparticulate. This study utilized the suggested dosing regimen published by PURIXAN^®^ to guide the dose volumes administered to study participants based upon their age and the 14 days was determined to be reflective of typical early chemotherapy oral treatment durations. Dosing and dispensing multiparticulate for pediatric formulations has never been done by direct consumers/users of the product. In most cases, applications for multiparticulate use have been limited to highly specialized pharmacies who create custom-made capsules in unit-dose formats. Note, this formulation type is restricted to unit-dose capsules and sachets. Sympfiny is the first system available to allow a caregiver to select a dose of the formulation and then dispense it into the child’s mouth thus delivered in the same means as liquid medication.

## 2. Materials and Methods

The materials used in this qualitative study of acceptability included the Sympfiny^®^ system and a placebo medication. Further details are provided below.

### 2.1. Sympfiny^®^ System

The Sympfiny^®^ system includes two main parts, an oral syringe and a bottle filled with multiparticulate (Figure 1A). The connection between the syringe and bottle (Figure 1B) allows both valves to open, initiating the flow of MP. Both valves close when the syringe is disconnected, stopping the flow of MP. The components of the syringe can be viewed in Figure 1C. This study was conducted with the 1 mL version of the Sympfiny^®^ syringe. Please refer to https://hs-design.com/Sympfiny/ (accessed on 14 November 2022) for Appendix A (instructions for use) and additional details.

### 2.2. Placebo

The placebo used for the study consisted of dry multiparticulate microspheres approximately 200 to 300 um in diameter. The microspheres are composed of microcrystalline cellulose with a spray coating moisture barrier consisting of known excipients. The placebo was produced in a pharmaceutical facility under General Manufacturing Process controls, additional information pertaining the multiparticulate can be viewed in Table 1.

PURIXAN’s dosing guidelines were used to convert liquid dosing volumes to the multiparticulate dosing volume based on the bulk density and excipient levels. The dose volumes administered to each age group are outlined in Table 2. This dosing schedule was intended to determine the acceptability of taking multiparticulate daily at an age-appropriate volume.

### 2.3. Participants

The sample size of study participants for acceptability testing is widely debated and dependent upon the methodology used and the ultimate endpoint. Since acceptability can be considered a type of sensory testing, guidance on sample size can be referenced from the Sensory Evaluation of Food–Principles and Practices which recommends a sample size of 25–40 participants when performing discrimination tests [9]. In addition, a previous acceptability study conducted with the Sympfiny^®^ 2 mL system utilized a sample size of 40 children (4–12 years old) [7].

This evaluation was conducted with a total of 60 children and 60 parents, representing approximately 15 child participants in each of the four age groups (1–2 years., 3–4 years., 5–8 years., 9–12 years.). Specific demographic information for the child and parent participants can be viewed in Table 3 and Table 4, respectively.

### 2.4. Methods

This study was conducted as a mix between two in-person sessions and fourteen days of at-home daily use. The initial session was held at HS Design’s (HSD) office and for the subsequent two weeks, participants used the Sympfiny^®^ system in their actual home environment and dispensed a daily dose of the multiparticulate placebo medication to the child. Participants returned to HSD’s office for the final in-person session.

The qualitative data collected during the in-person sessions was collected through observation of the parent and child’s usability of the device and observation of the child’s facial expression after taking the dose of MP. In addition, the moderator noted the parent and child’s respective subjective feedback of the process, device, and multiparticulate. A survey was made to collect the qualitative feedback during the at-home portion of the study.

The test moderator conducted the initial in-person session to introduce the parent and child participant to the Sympfiny^®^ system. Each parent and child were allowed to work as a team during this evaluation. Participants were given the system, IFU, and the indicated dose volume for their child’s age. Participants used the system to deliver 1 dose of the MP with guidance, as needed, from the moderator. While the participants were not instructed to take the dose with food or drink, water was available should the participant need it after taking the dose.

The parents were directed to a QR code that led to a survey on surveymonkey.com (accessed on 14 November 2022). The parents were instructed to fill out the survey with their corresponding participant ID number. All parent participants were instructed to fill out the first page of the survey, which asks about their experience with the Sympfiny system using a rating scale as described below. Additional subjective data could be entered as free text within the survey daily. If the child participant was four years old or younger, the parent participants were instructed to fill out the second page of the survey. Child participants between the ages five and twelve were instructed to fill out the second page of the survey themselves. The second page of the survey asked about the child’s experience with the multiparticulate. The survey page that the 5–12-year-old participants filled out, utilized a 5-point hedonic scale, as shown below in Figure 2. The parent participants survey also utilized a 5-point scale. The language of the scale was altered to accommodate the older audience, whereas 1–Very Difficult, 2–Difficult, 3–Neutral, 4–Easy, 5–Very Easy.

Before the end of the initial session, participants set the dosing clip on their second provided syringe and were provided instructions for the 14 day period of at-home use.

The participants returned to HSD’s office for a final in-person session at the end of the two weeks. They were asked a series of questions to understand how they adapted to using the Sympfiny^®^ 1 mL system. The participants were then asked to administer a final dose of the MP to the child and complete a final survey on their mobile phone or tablet. The session ended with the moderator asking participants to summarize their experience over the course of the study.

### 2.5. Data Collection

With informed consent, test personnel video recorded the initial and final in-person test session using a tripod-mounted digital video camera to document participant interactions, comments, and reactions.

During the in-person sessions, test personnel collected the following data about the parent participant: gender, use problems and potential root causes, subjective comments, participant responses to open-ended questions, responses to the first survey, and test personnel observations. The following data was recorded for the child participant: age, gender, facial expressions, behaviors prior to, during and after dose administration, visible residue remaining in their mouth, success of taking the sample (swallowed completely, partially, chewed on, spat out, stuck in throat, refused), number of syringe plunger pushes to administer a full dose, subjective comments and responses to open-ended and survey questions.

During the at-home portion of the study, participants filled out the same survey on surveymonkey.com (accessed on 14 November 2022) after each use. This data allowed the test personnel to track their experiences with the system and the MP. Participants’ survey responses were examined before the second in-person session. This allowed the study moderator to gain an understanding of the participants’ experience with the Sympfiny^®^ system and the MP. Knowing this, the study moderator was able to ask about days with less than favorable scores and gain an understanding of what happened on those specific days.

The following parameters were identified to indicate acceptability of the Sympfiny^®^ system: setting the dose, connecting the syringe, dispensing the multiparticulate, cleaning, accuracy of taking a dose. Child acceptability of the MP was indicated by facial expressions, mood/behavior, and comments. Children ages 5–12 were also asked to rate on a 5-point hedonic scale how they perceived the MP’s grittiness, volume, mouthfeel, and taste, as well answer if they were willing to the MP every day if they were sick. All survey questions can be viewed in Appendix B.

### 2.6. Data Analysis

After completing the second in-person session, participants’ survey results were imported into an excel sheet and the data was analyzed in 3 groups: parents, children aged 5–12, and children aged 4 and under. The survey ratings per day for each question were totaled and converted to percentages in order for the data to be comparable. This was completed by dividing each survey question’s total rating per day by the total number of responses collected for that day and multiplying that by 100. The percentages of each rating per day for each question were graphed using a bar graph.

Further analysis included asking the participants about particular days/survey questions that received a hedonic rating of 1–2. This was done to determine the root cause(s) of the negative rating to understand the attributes and any tasks/actions that influenced the negative rating.

## 3. Results

### 3.1. Child Participant Results

The survey, filled out by child participants aged 5–12, was used to make the following graphs. Children’s acceptability of the MP was evaluated based on data from the in-person sessions and the daily surveys. The participants rated the following categories using the hedonic scale: grittiness of multiparticulate, the volume of sample, mouthfeel of the sample, and overall taste.

#### 3.1.1. Overall Data for Ages 5–12

Figure 1 shows the average amount of ratings per day across all categories. The increase in favorable ratings over the two-week dosing period indicates that the majority of children had an adjustment period to the multiparticulate formulation and by the end of the two-week period were more accustomed to taking this form of medication.

Despite the approximate 30% of consistently low ratings seen in the above graph, approximately 90% of child participants aged 5–12 said they would be willing to take the medication in this formulation if they were sick (Figure 2).

#### 3.1.2. Grittiness of the Sample (Ages 5–12)

Figure 3 displays the daily responses in respect to the grittiness of the MP. The outlying data in day one could be attributed to the children being in an unfamiliar environment. The exact causality is undetermined. The increase of positive ratings suggest that the children acclimated to the grittiness of the MP.

#### 3.1.3. Volume of the Sample (Ages 5–12)

Children were asked to rate the volume, or amount, of MP they had to take. Based off the survey responses (Figure 4), approximately 60% of children indicated they liked the volume and an additional approximate 30% were neutral or had no opinion about the volume.

#### 3.1.4. Mouthfeel of Sample (Ages 5–12)

The survey results for how the children perceived the mouthfeel of the sample (Figure 5) shows that overtime the mouthfeel became more favorable. Participants who did not find the mouthfeel favorable stated in the final sessions that the MP stuck to their teeth and/or they needed water to wash it down.

#### 3.1.5. Overall Taste (Ages 5–12)

Figure 6 shows the ratings for overall taste of the MP. By the end of the two-week period, the ‘3—Not Sure’ ratings were minimal indicating a decision about the taste of the MP. Overall, the majority of children rated that they liked the taste of the multiparticulate.

#### 3.1.6. Willingness to Open Mouth to Receive Sample (Ages 1–4)

Parents of children aged 1–4 were asked through the daily survey if their children were willing to open their mouths for the syringe. Figure 7 below shows that the majority of participants opened their mouth for the MP; however, this number decreased with each daily dose. This may be due to the young participants’ expectations based off of prior days of receiving the dose.

#### 3.1.7. Did the Child Spit out the Placebo? (Ages 1–4)

Parents of children aged 1–4 also filled out if their child spat out the multiparticulate formulation. As seen in Figure 8, on average 12.6% of children ages 1–4 did spit out the placebo.

### 3.2. Parent Results

Parent’s acceptability of the Sympfiny^®^ system was evaluated based on data from the in-person sessions and the daily surveys. The participants rated the following categories using the hedonic scale: Setting the Dose, Connecting the Syringe, Dispensing the Multiparticulate, Accuracy of the Dose, and Cleaning. The following graphs were created from the at-home survey data.

#### 3.2.1. Parents: Overall Results

Although noting some discrepancy in opinion at the beginning of device use, overall, the parents had a favorable opinion of the Sympfiny^®^ system. Figure 9 below shows that on average approximately 95 percent of parents rated the system highly indicating the system’s acceptability.

Overall, the adult participants gave the device very high ratings across all categories which indicates that the device is acceptable. Through the daily survey, parents were asked to answer the question ‘Overall, do you consider this adapted oral syringe to be an acceptable device to administer multiparticulate to children?’ This was a free response question that the sixty adult participants answered every day for 14 days. On twelve occasions, adult participants did not submit a response to the question resulting in 828 possible answers. On 33 occasions, adult participants stated ‘No’ and there were 15 more occasions where the participants answers indicated that the device was not acceptable. According to those results, the device was not acceptable 5.8% of the time. 1.0% of the 5.8% participants that did not find the system acceptable was due to device malfunctions. The device malfunctions included the syringe breaking, the MP not distributing.

A total of three adult participants consistently answered ‘No’ to if the system is acceptable. All three participants were parents of children in the 1–4 age group and all three children responded negatively towards the MP.

#### 3.2.2. Setting the Dose

The Sympfiny^®^ system utilizes a dose clip to set the dose at a desired volume. As seen in Figure 10, the highest number of negative responses was reported on the first day.

The majority of negative responses are seen on day 1. Over the course of the two weeks, the negative ratings dramatically decreased. The few negative ratings in Figure 10 were attributed to the Sympfiny^®^ system not performing in the intended way. Participants recorded instances where the plunger rod snapped as the dose was being changed (Figure 3A) and where the dose clip broke (Figure 3B). The broken dose clip would not lock in place and therefore would not result in an accurate dosing volume.

#### 3.2.3. Connecting the Syringe

In order to dispense the MP, the syringe must be properly connected to the bottle. The task of connecting the syringe evaluated parents’ perception of connecting the syringe to the bottle (Figure 11). The majority of parents gave high ratings for this task. A few difficulties encountered by the parents included: making sure the syringe was fully connected, aligning the syringe to the bottle, and attempting to twist the syringe onto the bottle. Additionally, two participants broke the top of their syringes (Figure 4). The first break was caused by the plunger rod tip being exposed when firmly pressing down on the thumb rest while connecting the syringe (Figure 4A). This prevented the syringe from connecting and led to the plunger rod breaking. The other break was caused by the lateral force exerted on the plunger rod by the participants’ medial side of the palm while the participant was connecting the syringe to the bottle (Figure 4B).

#### 3.2.4. Dispensing the Multiparticulate

Dispensing the MP was rated very highly among parents (Figure 12). The minimal low ratings were associated with issues involving moisture getting into the bottle, the plunger rod becoming difficult to push, MP spilling out of the bottle, and having to tap the bottle excessively to get the MP out. The most common of these issues was the moisture getting into the bottle. The instructions that the participants received can be viewed in Appendix C. Please refer to Figure A1C, steps 16–18 for the proper way to clean the syringe. In summary though, participants were supposed to rinse the syringe, then disconnect the plunger rod from the barrel to air dry. If the syringe was not properly laid out to dry after cleaning, there would be residual moisture in the syringe at the time of the next dose. When connected to the bottle, the moisture from the syringe would seep into the bottle and cause clumps of MP that would inhibit the flow into the syringe.

#### 3.2.5. Cleaning

The majority of parents rated cleaning the Sympfiny^®^ syringe positively (Figure 13). Lower ratings were attributed to moisture in the bottle, residual MP on the bottle and the syringe and being unsure if the mouthpiece was cleaned due to its opaqueness. There is no data for day 1 as parents did not clean the syringe during the initial in-person session. Further, the instructions for how to clean the syringe can be seen in Appendix C.

#### 3.2.6. Accuracy of Dose

According to Figure 14 below, the majority of parents had confidence in delivering an accurate dose of the MP to their child. Doubts in the accuracy of the dose are due to moisture in the bottle, residual MP on the syringe’s mouthpiece and on the child’s mouth after delivering the dose, and MP spilling out of the syringe.

## 4. Discussion

This study was designed to test the acceptability of a multiparticulate formulation and the Sympfiny^®^ system by children and adult participants, respectively.

The older children (ages 5–12) were asked to rate the multiparticulate, across four categories: grittiness of multiparticulate, the volume of sample, mouthfeel of the sample, and overall taste. High ratings across all four categories increased by the end of the 14-day period. While there were lower ratings, approximately 90% of children ages 5–12 stated they would take the medicine if they were sick. The children who indicated that they would not take the medication even if they were sick, averaged 27.5% of the 1 ratings and 20.6% of the 2 ratings. Though an approximate 10% said they would not take the medication if they were sick, they were still able to successfully take the dose for 14 days. Per the EMA’s guideline stating the acceptability criteria is left to the applicant [1], the fact that the children were still able to take the dose of medication could indicate the MP formulation’s acceptability. If defining acceptability by success of taking the MP, then it can be determined that the formulation is acceptable for children ages 5 and over.

The parent participants of children ages 1–4 were asked to evaluate the child before, during, and after the dose and fill out the survey detailing the child’s response to the multiparticulate. On average, 12.6% of children (aged 1–4) spat out the dose every day as the study progressed. In addition, the number of children (aged 1–4) not willing to open their mouths for the syringe increased. Due to voluntary acceptance being an indicator of acceptability [2], the that the MP formulation may not be acceptable for long-term use for children 4 years-old or younger.

Adult participants were asked to rate five areas related to using the device: Setting the Dose, Connecting the Syringe, Dispensing the Multiparticulate, Accuracy of the Dose, and Cleaning. Overall, adult participants rated these categories very highly. The few low ratings were analyzed and a root cause was determined. Excluding instances of device failure, adult participants found the Sympfiny system to be acceptable 95.2% of the time. Of adults who did not rate the device as acceptable, 100% were parents of young children (ages 1–4) who displayed negative emotions towards the MP.

All survey answers are subjected to the participants interpretation of what the questions was asking [10]. Through the second in-person session, it was uncovered that adult participants interpreted the dispensing the multiparticulate question one of two ways: (1) dispensing the MP into the syringe or (2) dispensing into the child’s mouth. Should the study be conducted again, clarification around this question should be provided. In addition, study results are limited in determining causality of specific reasoning of ratings in that the free text data received was either missing or scant in description.

The design of the Sympfiny^®^ syringe and bottle system worked as intended. Potential suggested improvements include strengthening the plunger rod to avoid breakage during connection with the bottle. This could include additional feedback to the user that the connection to the bottle is secure. Overall these suggestions are minor and the device users were successfully able to dispense the prescribed dose as intended by design.

## 5. Conclusions

The data collected from the adult participants indicate that the Sympfiny^®^ syringe and bottle system is an acceptable delivery mode of multiparticulate formulations to children. The data collected from children ages 5–12 suggests that the multiparticulate formulation may have an adjustment period but is overall an acceptable form of medication. Approximately, 80% of children ages 1–4 were able to successfully take the dose of MP however, overtime the children became increasingly reluctant to open their mouths to receive the MP.

## Data Availability

Data is available for review and archieved within a secure server at HS Design.

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
