# Peer review of "Acceptability of Multiparticulate Dosing Using Sympfiny^®^ Delivery System with Children (Age 1–12)"

_pharmaceutics, 2022, doi:10.3390/pharmaceutics14112524_

Round 1
Reviewer 1 Report
This manuscript describes the investigation of the acceptability of multiparticulate dosing with Sympfiny in children. I think the data is clearly presented and the manuscript is well organized. On the other hand, there was a tendency to list the results, and the overall impression was that there was little discussion. I would like to make the following comments.
(1) Why does Rating 1 tend to increase from Day 4 to Day 6 for Grittiness of the multipaticulate (Graph 3) and Mouthfeel of the sample (Graph 5)?
(2) Insufficient description of scientific novelty.
(3) Insufficient identification of issues to be solved. Please describe what background issues exist and what you want to solve with this study.
(4) Based on the results of this study, do you have any comments on Sympfiny system (e.g., points that should be improved or functions that should be added)?
Author Response
- The study results are limited in determining the causality of specific reasoning of ratings in the free text data received was either missing or scant in the description. This limitation was added to the discussion.
-
The following was added to the introduction to address the novelty of the device: Dosing and dispensing multiparticulate for pediatric formulations has never been done by direct consumers/users of the product. In most cases, applications for multiparticulate use have been limited to highly specialized pharmacies who create custom-made capsules in unit-dose formats. Note, this formulation type is restricted to unit-dose capsules and sachets. Sympfiny is the first system available to allow a caregiver to select a dose of the formulation and then dispense it into the child’s mouth thus delivered in the same means as liquid medication.
- The following was added to discuss the primary objective and the issue to be solved: The primary objective of this paper is to evaluate whether it is acceptable to dose and dispense taste-masked multiparticulate drugs directly to a child using a designed system.
- The following was added to the discussion to address improvements of the device:
The design of the Sympfiny® syringe and bottle system worked as intended. Potential suggested improvements include strengthening the plunger rod to avoid breakage during connection with the bottle. This could include additional feedback to the user that the connection to the bottle is secure. Overall these suggestions are minor and the device users were successfully able to dispense the prescribed dose as intended by design.
Reviewer 2 Report
This is a clearly written well described study that should be published
I have some minor queries that may help clarify aspects of this paper.
Figure 2 shows phrasing from dislike very much yet the wording in the text line 156 is different - please clarify the wording used.
In the methods it would be useful to describe how the dose was taken, specifically with reference to co-administration with food or drink or the use of a drink or food immediately after dosing. This is of particular interest in line 248-249
In Graphs 1 and 2 was there a correlation between the low scores that those unwilling to take if they were sick?
The cleaning process for the device is not described and this may be useful as it is mentioned in lines 353-357 and section 3.1.3
Author Response
- The phrasing used in the paragraph describes the language of the adult hedonic scale survey. The language has been updated to make that clearer.
- The participants were not advised whether or not to take the dose with food or drink. There was water available for the participants after they received/took the dose, addressed in the methods section.
- The children who responded ‘no’ to if they would take the medication if they were sick, accounted for 27.5% of the ‘1’ ratings and 20.6% of the ‘2’ ratings. This information was added to the discussion.
- The intended cleaning procedure was added in section 3.2.4.
Round 2
Reviewer 1 Report
The authors appropriately respond to reviewers’ comments and suggestions.
I think this manuscript is now acceptable for publication.
Author Response
Thank you for reviewing our article. Your feedback was well received and helped clarify and improve the article!